Origin of aromatase inhibitory activity via proteochemometric modeling

Simeon Saw 1
Spjuth Ola 2
Lapins Maris 2
Nabu Sunanta 1
Anuwongcharoen Nuttapat 1 3
Prachayasittikul Virapong 3
Wikberg Jarl E.S. 2
Nantasenamat Chanin 1 chanin.nan@mahidol.ac.th
1 Center of Data Mining and Biomedical Informatics, Faculty of Medical Technology, Mahidol University , Bangkok , Thailand
2 Department of Pharmaceutical Biosciences, Uppsala University , Uppsala , Sweden
3 Department of Clinical Microbiology and Applied Technology, Faculty of Medical Technology, Mahidol University , Bangkok , Thailand
Sanderson J. Thomas
Electronic publication date: 2016 May 12
Publication date: 2016
Volume: 4
Electronic Location ID: e1979
Received 2015 Nov 26; Accepted 2016 Apr 6
Copyright: ©2016 Simeon et al.
Copyright year: 2016
Copyright holder: Simeon et al.
License: This is an open access article distributed under the terms of the Creative Commons Attribution License, which permits unrestricted use, distribution, reproduction and adaptation in any medium and for any purpose provided that it is properly attributed. For attribution, the original author(s), title, publication source (PeerJ) and either DOI or URL of the article must be cited.
License URL: https://creativecommons.org/licenses/by/4.0/

Keywords: Aromatase, Quantitative structure–activity relationship, Breast cancer, Data mining, QSAR, Aromatase inhibitor, Proteochemometrics

Funding: Swedish Research Links program C0610701 National Research University Initiative JESW and CN are supported by a joint grant from the Swedish Research Links program (no. C0610701) from the Swedish Research Council. This work is also partially supported by the Office of Higher Education Commission and Mahidol University under the National Research University Initiative. The funders had no role in study design, data collection and analysis, decision to publish, or preparation of the manuscript.

==============================
Aromatase, the rate-limiting enzyme that catalyzes the conversion of androgen to estrogen, plays an essential role in the development of estrogen-dependent breast cancer. Side effects due to aromatase inhibitors (AIs) necessitate the pursuit of novel inhibitor candidates with high selectivity, lower toxicity and increased potency. Designing a novel therapeutic agent against aromatase could be achieved computationally by means of ligand-based and structure-based methods. For over a decade, we have utilized both approaches to design potential AIs for which quantitative structure–activity relationships and molecular docking were used to explore inhibitory mechanisms of AIs towards aromatase. However, such approaches do not consider the effects that aromatase variants have on different AIs. In this study, proteochemometrics modeling was applied to analyze the interaction space between AIs and aromatase variants as a function of their substructural and amino acid features. Good predictive performance was achieved, as rigorously verified by 10-fold cross-validation, external validation, leave-one-compound-out cross-validation, leave-one-protein-out cross-validation and Y-scrambling tests. The investigations presented herein provide important insights into the mechanisms of aromatase inhibitory activity that could aid in the design of novel potent AIs as breast cancer therapeutic agents.

Introduction

Cancer exerts a great impact on the quality of life of patients and is the leading cause of death worldwide. Breast cancer is the most common cancer type and is the second most common cause of death in women worldwide (Fontham et al., 2009). Despite the continuous efforts being made towards improving diagnostic tests, the incidence rate of breast cancer has gradually increased (May, 2014). It is estimated that around two-thirds of breast cancers in women are dependent on the steroid hormone estrogen, which regulates tumor cell growth and drives the progression of the cancer (Lipton et al., 1992). Therefore, two major therapeutic approaches are involved in breast cancer treatment and prevention: the first involves the development of drugs that target the estrogen receptor, which are also known as selective estrogen receptor modulators (SERMs), whereas the second approach involves the development of drugs that target aromatase (i.e., the enzyme that converts androgens to estrogens) and are thus known as aromatase inhibitors (AIs).

Aromatase, also known as cytochrome P450 19A1 (EC 1.14.14.1), is the expression product of the CYP19A1 gene. The enzyme comprises 503 amino acids spanning twelve α-helices and ten β-strands, inside which sits a heme co-factor that is coordinated by a cysteine residue at position 437 (Ghosh et al., 2009). Aromatase is a major producer of estrogen in post-menopausal women, and it catalyzes the rate-limiting step converting androgens to estrogens (Simpson et al., 1994). The aromatase conversion of androgens to estrogens involves three steps, whereby androgen’s methyl group at carbon 19 is oxidized to form formic acid, which is followed by the aromatization of the A ring to the phenolic A ring of estrogen (Eisen et al., 2008). As aromatase catalyzes the biosynthesis of estrogen from androgens, inhibition of aromatase activity has become the standard treatment for hormone-dependent breast cancers in women.

In general, aromatase inhibitors can be classified into two major types according to their chemical structures, steroids and non-steroids inhibitors. The steroidal inhibitors are also known as mechanism-based inhibitors, as they bind covalently to aromatase, thus destroying the enzymes by forming irreversible interactions. On the other hand, non-steroidal inhibitors have reversible inhibitory interactions with the heme co-factor of aromatase. Although aminoglutethimide, a first generation non-steroidal inhibitor, can inhibit the action of aromatase while also affording poor selectivity as it can also inhibit other cytochrome P450 enzymes involved in the biosynthesis of cortisol aldosterone thereby leading to severe side effects. Because of these side effects, aminoglutethimide was withdrawn from clinical use. The second-generation aromatase inhibitors consisted of fadrozole and formestane, which are non-steroidal imidazole derivatives and steroidal analogs, respectively. Although fadrozole was more selective and potent than aminoglutethimide, it still has undesirable effects, including inhibitory action against the production of aldosterone, corticosterone and progesterone. Formestane was the first aromatase to be used clinically, but the effects of covalently binding to aromatase led to its name of suicide inhibitor. The third-generation non-steroidal aromatase inhibitors include vorozole, anastrozole and letrozole; it should be noted that the latter two are marketed under the trade names of Arimidex and Femara, respectively. The current standard-of-care compounds for preventing relapse of breast tumors are anastrozole, letrozole and exemestane (Ma et al., 2015). However, in the early and advanced stages of breast cancer, 20% of patients suffer relapse of the disease ( Early Breast Cancer Trialists’ Collaborative Group (EBCTCG) et al., 2011), and the disease eventually progress despite AI therapy, leading to the disease becoming incurable, lethal and systemic. The mechanisms of aromatase resistance are heterogeneous, and the hallmarks range from changes in the tumor microenvironment, deregulation of the ER pathway, decrease in apoptosis and senescence, abnormality in the cell cycle machinery, increase in cancer stem cells, overexpression of EGFR in the growth factor receptor pathway and mutations in PIK3CA, PTEN and AKT1 through secondary messengers (Ma et al., 2015).

Previously, our group utilized the quantitative structure-activity relationship (QSAR) method in our efforts to understand the origin of aromatase inhibition (Nantasenamat et al., 2013a; Nantasenamat et al., 2013b; Worachartcheewan et al., 2014a; Worachartcheewan et al., 2014b; Nantasenamat et al., 2014; Shoombuatong et al., 2015). We also used structure-based approaches to elucidate how selected compounds of interest interact with aromatase to give rise to their inhibitory activity (Suvannang et al., 2011; Worachartcheewan et al., 2014b; Pingaew et al., 2015). Although robust, both ligand-based and structure-based approaches have limitations: the former will only allow the study of how modifications to functional moieties of ligands influence the bioactivity, whereas the latter will only provide insights into how the spatial location of amino acid residues influences the bioactivity.

In this study, we developed a unified proteochemometric (PCM) model to investigate the interaction between a series of ligands and a series of aromatase variants. Such computational approaches present methodological differences with the systems-based approach (i.e., the PCM model) described herein. To this end, aromatase protein variants were represented using highly interpretable and position-specific z-scale descriptors, while AIs were represented using substructure fingerprint descriptors. Each interacting pair of AIs with aromatase variants was assigned a pIC50 value. Various machine learning methods were then employed to model the interaction between the ligands and the aromatase variants. Compared to the conventional ligand-based QSAR approach, the PCM technique represents a leap forward for structure–activity relationship investigations due to its ability to simultaneously consider descriptive information of several proteins and several ligands as well as its inherent interpretability in which the relative significance of descriptors in relation to the dependent variable (i.e., pIC50) can be derived. Furthermore, such PCM strategies provide important insights into the molecular basis for the inhibition of a set of AIs against a set of aromatase variants and may aid in the combat against aromatase inhibitor resistance.

Figure 1 Workflow for PCM modeling of aromatase inhibitory activity.

Figure 2 Chemical structures of aromatase inhibitors.

Materials & Methods

Data set

A data set of compounds, site-specific variations of residues, and bioactivity values for protein-compound pairs was obtained from previous studies by Kao et al. (1996) and Auvray et al. (2002). The general workflow for PCM modeling of this data set is summarized in Fig. 1. The compounds included in this study are comprised of 4-hydroxyandrostenedione (4-OHA, 1), bridge (2,19-methyleneoxy) androstene-3,17-dione (MDL 101003, 2), 7α-(4′-amino)phenylthio-1,4-androstandiene-3,17-dione (7α-APTADD, 3), aminoglutethimide (4), letrozole (5), vorozole (6), anastrozole (7), MR20814 (8), MR20492 (9) and MR20494 (10). The chemical structures of these compounds are shown in Fig. 2. These compounds interact with target proteins to induce pharmacological effects. However, the interaction occurs at the active site, where the compounds bind to only a small portion of residues in the target proteins. However, residues that are involved both near and far way from the active site can be considered in the PCM model. In this study, residues K119, C124, K130, I133, F235, E302, P308, D309, T310, F320, I395, I474 and D476 were considered. These residues cover the AI binding site as well as residues near the aromatase active site. Aromatase inhibitory activities were originally defined using IC50 values, but to obtain a more distributed spread of the data points, they were subjected to negative logarithmic transformation, yielding pIC50 values. A summary table of the pIC50 values for each pair of aromatase variant and compound is provided in Data S1.

Compound descriptors

The chemical structures of the compounds were drawn using Marvin Sketch version 6.2.1 (ChemAxon Ltd., 2014) and subsequently pre-processed according to the QSAR data curation workflow described by Fourches, Muratov & Tropsha (2010). In the workflow, metal ion containing compounds were removed because reliable descriptors cannot be calculated when compounds contain metal ions. The second part involved removing any counter ions from the compounds, followed by the normalization of the chemotypes and standardization of tautomers using the built-in function of the software program PaDEL-Descriptor (Yap, 2011). The curated compounds were subsequently coded using substructure fingerprint counts (Laggner, 2009). Fingerprint descriptors are numerical values that are used to describe the structure of compounds, including the number of hydroxyl groups and the number of benzene rings. In particular, substructure fingerprints were chosen to describe the compounds because they are interpretable and can therefore pinpoint the substructures in compounds that are important for inhibiting aromatase.

Protein descriptors

Aromatase comprises a polypeptide chain of 503 amino-acid residues and a prosthetic heme group at its active site. An androgen-specific cleft, consisting of hydrophobic and polar residues, is situated at the substrate binding site (Simpson et al., 1994). Of the 503 amino acids, 13 amino acid positions were found to be mutated in the investigated variants, as shown in Fig. 3. Each of the amino acid positions was encoded using a set of three z-scale descriptors, thus giving 39 z-scale descriptors for each of the 22 aromatase proteins. z-scale descriptors characterize the 20 naturally occurring amino acids by encapsulating 29 physicochemical descriptors, comprising 9 experimentally determined values for retention times in thin-layer chromatography, 7 nuclear magnetic resonance shift values, 2 pK values of amino acids from amino groups and carboxylic acid groups, van der Waals volume, MW, isoelectric point, paper chromatography value, dG of the transfer of amino acids, hydration potential, salt chromatography value, and log P, log D and dG of accessible amino acids along three principal components. This high-dimensional set of values is reduced to a low-dimensional set of variables using principal component analysis, giving rise to a set of 3 z-scale descriptors, where z1 essentially represents the hydrophobicity/hydrophilicity, z2 represents the side-chain bulk volume, and z3 represents the polarizability and charge of the amino acids (Hellberg et al., 1987).

Figure 3 Three-dimensional structure of aromatase and investigated sites of mutations.

The protein structure (PDB 3S79) and mutated residues are shown in cartoon representation and ball-and-sticks, respectively, where both are colored by their secondary structures (α helices are colored cyan while β strands are colored purple). The natural substrate androstenedione and the heme prosthetic group are shown as orange colored ball-and-sticks.

Data partitioning

The K-means clustering algorithm was used to partition the data into two groups, the internal and external sets. The algorithm selects a set of cluster centers to start the K-means clustering directly in Euclidean space whereby samples closest to the center cluster are picked from each cluster. The naes function prospectr from the R package was used to split the data; 80% of the protein-ligand pairs were used as the internal set and the remaining 20% were used as the external set (Stevens & Ramirez-Lopez, 2013).

Feature selection

Intercorrelation, also known as collinearity, is a condition in which pairs of descriptors are known to have substantial correlations. Because it adds more complexity to models than the information they provide and also could potentially give rise to bias, it therefore has a negative impact on PCM analysis. Thus, the cor function from the R package stats (R Core Team, 2014) was used to calculate the pairwise correlation between descriptors, and a descriptor in a pair with a Pearson’s correlation coefficient greater than the threshold of 0.7 was filtered out using the findCorrelation function with the cutoff set at 0.7 from the R package caret so as to obtain a smaller subset of descriptors (Kuhn, 2008).

Principal component analysis

Principal component analysis (PCA) is a widely used method for finding the linear combination of a set of observations with the most possible variance, and it can reveal important characteristics of the data structures, which are otherwise difficult to distinguish. To obtain the optimal number of PCs, Horn’s parallel analysis was applied to the biological space of aromatase variants (Zwick & Velicer, 1986). To allow comparisons, the same number of PCs as that obtained from Horn’s parallel analysis of aromatase variants was used also for the chemical space of AIs. Four PCs were deemed as sufficient for providing meaningful information on the chemical space of both AIs and aromatase variants. PCA was performed using the R statistical programming language. Descriptors with a variance close to zero were removed using the nearZeroVar function of the R package caret (Kuhn, 2008). The prcomp and kmeans functions from the R package stats were used to perform PCA and K-Means clustering, respectively (R Core Team, 2014). Prior to PCA analysis, all the data were centered and scaled to have a unit variance using the center and scale functions. The paran function with the argument for the iterations set as 5,000 from the R package paran was utilized to perform Horn’s parallel analysis to determine the optimal number of PCs (Dinno, 2012). Plots were created using the R package ggplot2 with a 95% confidence ellipse drawn around the clusters (Wickham, 2009) as shown in Fig. 4.

Compound-protein cross-terms

The goal of PCM analysis is to relate the compound and target spaces with the interaction activity by creating a mathematical representation of the interaction space. Thus, unlike QSAR in which the compounds’ chemical spaces are independently related to biological activities, PCM links the unified compounds and protein space to represent their ability to form non-covalent interactions. In addition to compound descriptors and protein descriptors, PCM also makes use of cross-terms as a representation of interactions between compounds and proteins. In this study, cross-terms were calculated as the mathematical product of the compounds descriptors with those of the protein descriptors. Cross-terms were computed using the getCPI function from the R package Rcpi (Cao et al., 2014). Moreover, the total number of cross-terms computed for self interaction (i.e., compound × compound and protein × protein) was obtained as follows: (1) NN−12

where N is the total number of descriptors of compounds or proteins.

Multivariate analysis

Descriptors of the chemical compounds and investigated amino acids residues were used for constructing models for predicting the pIC50 activity using several machine learning methods.

Random forest (RF) is an ensemble classifier that comprises multiple decision tress. Decision trees are powerful and transparent classifiers, which use a tree structure to model the relationship between the descriptors and the classes. Optimal tuning parameters (i.e., mtry) for RF were obtained by training the model with different ranges of mtry accompanied with 10-fold cross-validation. The train function from caret was used with the argument trControl set as 10-fold cross-validation with 100 iterations. The randomForest function from the R package randomForest was used to build the predictive models with 500 decision tress (Liaw & Wiener, 2002). To avoid the possibility of chance correlation that may arise from random seed of a single data partition, the models were built from 100 independent data partitions as described above using K-means clustering.

Partial least squares (PLS) is an extension of PCA that correlates the X matrix of predictors with the Y dependent variables by simultaneously projecting X onto the latent variables and finding linear relationships between them. The PLS model was fine-tuned with the train function from the caret package, and this operator was used to extract the optimal number of PCs for building the predictive model. Finally, the plsr function from the R package pls was used to build PLS models with different combinations of predictors (Mevik & Wehrens, 2007).

Ridge regression (RR) is effective at reducing the predictive model variance by minimizing the residual sum of squares. This is done by dividing the values of all the descriptors by their variance. Ridge regression was performed using linearRidge from the R package ridge. The parameter for the model was fine-tuned with the train function from the R package caret.

Support vector machine (SVM) is a statistical learning method that is widely popular owing to its robust performance. However, the downside of the approach is its black box nature in that it can be difficult to understand its inner workings. SVM learns by finding the maximal hyperplane to differentiate data points in the feature space. Prior to discerning of the aforementioned complex relationship of the X and Y vectors by means of the maximal hyperplane, the original input space must first be mapped onto a higher dimensional space by means of the kernel function. The svm function from e1071 R package was used to train the SVM models using radial basis kernel (Meyer et al., 2008). The model were fine tuned with tune function from the e1071 R package to obtain optimal parameters (i.e., the γ and C parameters) (Meyer et al., 2008).

Bayesian linear regression (BLR) correlates the bioactivity of interest using the distribution of parameters with a set of values sampled from the posterior distribution and averaged to derive the final prediction. The posterior probabilities are typically obtained through Monte Carlo simulations to subsequently derive the regression coefficients. The bayesglm function from the arm R package was used to build the BLR models (Gelman et al., 2009).

To avoid the possibility of chance correlation afforded by biased data sampling, the data set was subjected to 100 independent 80/20 data splits to internal and external data subsets. Models were then built for each data split and the statistical assessment parameters (i.e., R2, Q2 and RMSE) were reported as the mean and standard deviation values calculated from the 100 models.

Validation of model performance

The internal validation set (i.e., the 80% data subset) was subjected to training and 10-fold cross-validation (10-fold CV). In a 10-fold CV scheme, one fold of the data was left out as the testing set, while the remaining were used as the training set for building the predictive model. This was repeated iteratively until all folds were left out once. The defaultSummary function from the R package caret was used to obtain statistical assessment parameters for validating the PCM models Kuhn (2008). The external set was then used to validate the predictability of the constructed PCM models, and the goodness-of-fit (R2), predictive ability (Q2) and root mean squared error (RMSE) were determined.

In addition, leave-one-protein-out (LOPO) cross-validation and leave-one-compound-out (LOCO) cross-validation were also used to externally validate the PCM models for their extrapolation abilities in terms of new proteins or compounds. In the LOPO scheme, data annotated for single protein are left out as the test set while the remaining data are used to build the predictive model. Similarly, in the LOCO scheme, one compound is iteratively left out as the test set and evaluated against the trained model. Both processes were repeated iteratively until each aromatase variant and compounds had a chance to be left out as the test set.

To assess the statistical significance of R2 and Q2, the Y-scrambling test, a well-established statistical method also known as permutation testing, was used to ensure the robustness of the PCM models to rule out the possibility of chance correlations or redundant data sets. In the test, the true Y-dependent variable is randomly shuffled, and the statistical assessment parameters are recalculated. The permute function from the R package gtools was used to scramble the Y-dependent variables (i.e., pIC50) (Warnes, Bolker & Lumley, 2015).

Results and Discussion

Biological and chemical space of aromatase variants and compounds

PCA was utilized to analyze the z-scale descriptors of the aromatase variants for a better understanding of the biological space. Horn’s parallel analysis deemed four PCs sufficient to yield information for satisfactorily explaining the biological space. The overall percentage of the total explained variance of the first four PCs was 75.02%, which is indicative of the good coverage of the data modeled by these PCs.

PC1 accounted for 22.07% of the data variance, in which the positive ends were dominated by p133z2 (side-chain bulk volume of the amino acid at position 133 of the aromatase variants), p133z3 (polarizability and charge of the amino acid at position 133 of the aromatase variants), and p133z1 (hydrophobicity/hydrophilicity of the amino acid at position 133 of the aromatase variants), whereas p474z3 (polarizability of the amino acid at position 474 of the aromatase variants), p474z2 (side-chain bulk volume of the amino acid at position 474 of the aromatase variants), p476z3 (polarizability and charge of the amino acid at position 476 of the aromatase variants), p476z1 (hydrophobicity/hydrophilicity of the amino acid at position 476 of the aromatase variants) and p474z1 (hydrophobicity/hydrophilicity of the amino acid at position 474 of the aromatase variants) had high loadings for the negative ends. It can be observed that the physicochemical properties of position 133 have a strong influence, as they provide high loadings on one side, whereas the physiochemical properties of position 474 account for high loadings on the other side. The descriptors p119z3 (polarizability and charge of the amino acid at position 119) and p119z2 (side-chain bulk volume of the amino acid at position 119) did not provide much variance for PC1.

PC2 explained 21.21% of the variance for the protein descriptors. The descriptors with the highest loadings were p474z3 (polarizability and charge of the amino acid at position 474 of the aromatase variants), p474z2 (side-chain bulk volume of the amino acid at position 474 of the aromatase variants) and p474z1 (hydrophobicity/hydrophilicity of the amino acid at position 474 of the aromatase variants) for the positive ends, while the negative ends were dominated by p133z2 (side-chain bulk volume of the amino acid at position 133 of the aromatase variants), p133z3 (polarizability and charge of the amino acid at position 133 of the aromatase variants), p476z3 (polarizability and charge of the amino acid at position 476 of the aromatase variants) and p476z1 (hydrophobicity/hydrophilicity of the amino acid at position 476 of the aromatase variants).

PC3 accounted for 20.04% of the data variation. It can be observed that PC1 and PC2 have the same explained variance as PC3, accounting for a total explained variance of 63.31%. For PC3, the descriptor providing the highest loadings for the positive end was p119z3 (polarizability and charge of the amino acid at position 119 of the aromatase variants), whereas p199z1 (hydrophobicity/hydrophilicity of the amino acid at position 119 of the aromatase variants), p119z2 (side-chain bulk volume of the amino acid at position 119 of the aromatase variants) and p113z2 (side-chain bulk volume of the amino acid at position 113 of the aromatase variants) and p113z3 (polarizability and charge of the amino acid at position 113 of the aromatase variants) had a large influence on the negative ends.

PC4 accounted for 11.70% of the explained variance. For PC4, the descriptors with high loadings for the positive side were p474z3 (polarizability and charge of the amino acid at position 474 of the aromatase variants) and p474z2 (side-chain bulk volume of the amino acid at position 474 of the aromatase variants), whereas p119z1 (hydrophobicity/ hydrophilicity of the amino acid at position 119 of the aromatase variants) and p119z2 (side-chain bulk volume of the amino acid at position 119) had the highest loadings for the negative side.

For comparative purpose with PCA analysis of the biological space of aromatase variants, 4 PCs were also used for the PCA analysis of the chemical space of AIs, which made use of substructure fingerprint descriptors. The cumulative proportion of the explained variance of the first 4 PCs was 81.22%, which can seem to provide enough information for insights on the data, as the data appear geometrical in the feature space. PC1 accounted for 38.89% of the data variance. It can be noted that the first PC was the most informative, as it explained the highest data variation among the PCs. It can be observed that the highest descriptor effects of PC1 were SubFPC49 (ketone), SubFPC300 (1,3-tautomerizable), SubFPC301 (1,5-tautomerizable), SubFPC4 (quaternary carbon), SubFP2 (secondary carbon) and SubFPC3 (tertiary carbon) on one end, while the other end was dominated by SubFPC295 (C ONS bond), SubFPC184 (heteroaromatic), SubFPC181 (hetero N nonbasic), SubFPC275 (heterocyclic) and SubFPC302 (rotatable bond). SubFPC12 (alcohol), SubFPC76 (enamine), SubFPC135 (vinylogous carbonyl or carboxyl derivative) and SubFPC13 (primary alcohol) had low loadings on PC1, suggesting that they only provide low data variation in terms of AI. It can be seen that in substructures, chemical conjugation, a phenomenon in which p-orbitals are connected, thereby allowing electrons to flow within the conjugated system, provided the highest afforded loadings in PC1.

PC2 accounted for 18.45% of the data variance, and descriptors providing the high loading on the positive ends were SubFPC1 (primary carbon), SubFPC35 (ammonium), SubFPC134 (isonitrile), SubFPC296 (charged), SubFPC297 (anion), SubFPC298 (cation) and SubFPC299 (salt), whereas SubFPC287 (conjugated double bond), SubFPC13 (primary alcohol), SubFPC12 (alcohol), SubFPC76 (enamine) and SubFPC135 (vinylogous carbonyl or carboxyl derivative) dominated the negative ends. Interestingly, the substructures associated with charge showed the most variance in describing the data variation at PC2. In contrast, SubFPC49 (ketone), SubFPC5 (alkene) and SubFPC275 (heterocyclic) provided little information.

PC3 accounted for 12.63% of the data variance for AI. PC3 thus represented just a small proportion of the data variance compared with the lower-order PCs. However, the spread of the data for PC3 was sufficiently large for it to be viewed as informative. The loadings of PC3 mainly comprised SubFPC13 (primary alcohol), SubFP12 (alcohol), SubFPC76 (enamine) and SubFPC135 (vinylogous carbonyl or carboxyl derivative) on the positive ends, whereas SubFPC307 (chiral center specified), SubFPC5 (alkene), SubFPC171 (arylchloride) and SubFPC180 (hetero N basic no H) dominated the negative ends.

PC4 had an explained variance of 11.25%. The descriptors that capture high loadings at the positive end were SubFPC20 (alkylarylthioether), SubFPC38 (alkylarylthioether), SubFPC96 (carbodithioic ester), SubFPC137 (vinylogous ester) and SubFPC303 (Michael acceptor). In contrast, the negative ends were dominated by SubFPC88 (carboxylic acid derivative), SubFPC105 (imide acidic), SubFPC171 (arylchloride), SubFPC275 (heterocyclic) and SubFPC72 (enol).

A closer look at the data structures for both chemical descriptors and protein descriptors revealed that the chemical descriptors provided better systemic data types when compared to the protein descriptors. It can be observed that of the overall explained variance of the first two PCs, 57.34% and 43.28% were accounted for by compound and protein descriptors, respectively. Thus, in comparison, it can be concluded that the compound descriptors represent data structures with more useful information, whereas the protein descriptors contain noise in the data. Noise in the data structure may just add to the complexity of the model, causing overfitting and thereby producing unstable models. Nevertheless, the first four PCs afforded overall variance in the data of 81.22%, and 75.02% for compounds and proteins, respectively.

PCM modeling of aromatase inhibitory activity

PCM allows the study of ligand-protein interactions by simultaneously investigating the interaction of several compounds against several proteins (i.e., in this case several aromatase variants). Our earlier QSAR models of the inhibitory properties of AI used only information from chemical compounds while the potential effects of protein binding sites and residues on the inhibitory properties of AI were not considered. This study addresses this issue by applying PCM modeling to integrate information on the interaction space of both proteins and ligands into one unified model.

The approach seems rational in view of an earlier PCM investigation by Prusis et al. (2006), where the amino acid position located very far from the binding site of a peptide hormone receptor could be effectively studied via PCM. One of the biggest problems with PCM modeling is that the data matrix tends to be very large, which leads to a high computational cost and may be prone to overfitting. To remove irrelevant descriptors that contribute more noise to the model than the information they provide, therefore feature selection was performed by removing descriptors that have pairwise Pearson’s correlations higher than the cutoff threshold of 0.7. Such threshold was chosen because Pearson’s correlation coefficients that are larger in value are indicative of high collinearity between descriptors (Booth, Niccolucci & Schuster, 1994).

Several machine learning methods (e.g., RF, PLS, RR, SVM and BLR) were employed in the construction of PCM models using various combinations of descriptor blocks (Table 1) after which the best performing model will be identified and selected for further interpretation on the importance of underlying features. Table 2 summarizes the prediction performance in terms of the squared correlation coefficient values while Table 3 provides the RMSE values of the models. In general, RF afforded consistent and robust performance as supported by high squared correlation coefficient values (QCV2 in excess of 0.83) and low RMSE values (less than or equal to 0.50) for almost all models except for models 2 and 5 (i.e., comprising the P and P × P descriptor blocks, respectively). Results from performance comparison indicated the following trend for the squared correlation coefficients (in order of decreasing values): RF > SVM > PLS > RR > BLR. Furthermore, evaluation of the RMSE values from Table 3 also showed the identical but inverted order (in order of increasing values): RF < SVM < PLS < RR < BLR. It is also interesting to note that only RF afforded consistently high squared correlation coefficients and low RMSE values for most of its models while the other learning methods provided fluctuating levels of performance where some performed well while others did not.

Table 1 Number of descriptors in the 13 PCM models formed by different combination of the descriptor blocks.

Model	Number of descriptors	
	C	P	C × P	C × C	P × P	Total	
1	13	0	0	0	0	13	
2	0	18	0	0	0	18	
3	0	0	234	0	0	234	
4	13	18	0	78	0	78	
5	13	18	0	0	153	153	
6	13	18	0	0	0	31	
7	13	18	234	0	0	165	
8	13	18	0	78	0	109	
9	13	18	0	0	153	184	
10	13	18	234	78	0	343	
11	13	18	234	0	153	418	
12	13	18	0	78	153	262	
13	13	18	234	78	153	496	

Table 2 Summary of the squared correlation coefficients from the 13 PCM models constructed by various machine learning methods.

Model	RF	PLS	RR	SVM	BLR	
	RTr2	QCV2	QExt2	RTr2	QCV2	QExt2	RTr2	QCV2	QExt2	RTr2	QCV2	QExt2	RTr2	QCV2	QExt2	
1	0.87 ± 0.00	0.86 ± 0.12	0.93 ± 0.01	0.88 ± 0.01	0.86 ± 0.11	0.93 ± 0.01	0.88 ± 0.01	0.86 ± 0.11	0.93 ± 0.01	0.87 ± 0.01	0.85 ± 0.10	0.93 ± 0.01	0.87 ± 0.01	0.86 ± 0.10	0.92 ± 0.01	
2	0.35 ± 0.02	0.25 ± 0.22	0.25 ± 0.22	0.20 ± 0.02	0.16 ± 0.20	0.21 ± 0.11	0.34 ± 0.03	0.20 ± 0.23	0.17 ± 0.10	0.29 ± 0.03	0.23 ± 0.21	0.14 ± 0.08	0.34 ± 0.03	0.20 ± 0.20	0.16 ± 0.10	
3	0.95 ± 0.01	0.84 ± 0.14	0.90 ± 0.03	0.86 ± 0.02	0.61 ± 0.22	0.54 ± 0.12	0.96 ± 0.01	0.53 ± 0.26	0.63 ± 0.15	0.95 ± 0.02	0.75 ± 0.17	0.73 ± 0.10	0.96 ± 0.01	0.47 ± 0.28	0.50 ± 0.17	
4	0.88 ± 0.01	0.85 ± 0.12	0.93 ± 0.01	0.87 ± 0.05	0.86 ± 0.11	0.93 ± 0.01	0.87 ± 0.01	0.86 ± 0.11	0.93 ± 0.01	0.87 ± 0.01	0.85 ± 0.12	0.93 ± 0.01	0.87 ± 0.01	0.83 ± 0.15	0.93 ± 0.01	
5	0.32 ± 0.03	0.18 ± 0.19	0.33 ± 0.12	0.22 ± 0.02	0.18 ± 0.18	0.30 ± 0.13	0.35 ± 0.03	0.19 ± 0.21	0.33 ± 0.12	0.29 ± 0.03	0.14 ± 0.16	0.20 ± 0.11	0.35 ± 0.02	0.23 ± 0.20	0.34 ± 0.12	
6	0.93 ± 0.01	0.85 ± 0.11	0.90 ± 0.04	0.92 ± 0.01	0.87 ± 0.09	0.89 ± 0.04	0.93 ± 0.01	0.86 ± 0.10	0.87 ± 0.05	0.94 ± 0.01	0.84 ± 0.14	0.86 ± 0.06	0.93 ± 0.01	0.81 ± 0.18	0.87 ± 0.06	
7	0.96 ± 0.01	0.83 ± 0.15	0.89 ± 0.05	0.87 ± 0.01	0.63 ± 0.20	0.63 ± 0.16	0.91 ± 0.01	0.63 ± 0.23	0.62 ± 0.16	0.97 ± 0.01	0.79 ± 0.15	0.69 ± 0.09	0.96 ± 0.01	0.61 ± 0.25	0.54 ± 0.20	
8	0.96 ± 0.01	0.83 ± 0.15	0.89 ± 0.05	0.90 ± 0.01	0.81 ± 0.13	0.88 ± 0.06	0.90 ± 0.01	0.75 ± 0.16	0.82 ± 0.06	0.95 ± 0.02	0.84 ± 0.10	0.81 ± 0.07	0.92 ± 0.01	0.87 ± 0.09	0.86 ± 0.05	
9	0.96 ± 0.01	0.85 ± 0.12	0.89 ± 0.04	0.87 ± 0.01	0.72 ± 0.16	0.74 ± 0.08	0.74 ± 0.02	0.70 ± 0.15	0.64 ± 0.08	0.95 ± 0.02	0.81 ± 0.18	0.83 ± 0.06	0.92 ± 0.02	0.86 ± 0.08	0.86 ± 0.05	
10	0.96 ± 0.01	0.84 ± 0.15	0.90 ± 0.04	0.82 ± 0.01	0.62 ± 0.22	0.58 ± 0.13	0.93 ± 0.01	0.67 ± 0.24	0.63 ± 0.12	0.96 ± 0.01	0.78 ± 0.17	0.68 ± 0.14	0.95 ± 0.01	0.79 ± 0.24	0.66 ± 0.23	
11	0.96 ± 0.01	0.85 ± 0.12	0.88 ± 0.04	0.90 ± 0.01	0.72 ± 0.20	0.63 ± 0.12	0.78 ± 0.01	0.65 ± 0.24	0.62 ± 0.15	0.96 ± 0.01	0.67 ± 0.19	0.74 ± 0.08	0.94 ± 0.01	0.59 ± 0.30	0.67 ± 0.23	
12	0.94 ± 0.01	0.86 ± 0.11	0.90 ± 0.04	0.83 ± 0.01	0.72 ± 0.19	0.79 ± 0.09	0.91 ± 0.01	0.75 ± 0.18	0.82 ± 0.07	0.92 ± 0.04	0.71 ± 0.21	0.78 ± 0.13	0.91 ± 0.01	0.87 ± 0.10	0.86 ± 0.05	
13	0.94 ± 0.01	0.86 ± 0.11	0.90 ± 0.04	0.84 ± 0.01	0.74 ± 0.19	0.80 ± 0.07	0.84 ± 0.01	0.78 ± 0.18	0.83 ± 0.06	0.92 ± 0.04	0.69 ± 0.23	0.77 ± 0.11	0.91 ± 0.01	0.84 ± 0.16	0.87 ± 0.06	

Table 3 Summary of the root mean squared error from the 13 PCM models constructed by various machine learning methods.

Model	RF	PLS	RR	SVM	BLR	
	RMSETr	RMSECV	RMSEExt	RMSETr	RMSECV	RMSEExt	RMSETr	RMSECV	RMSEExt	RMSETr	RMSECV	RMSEExt	RMSETr	RMSECV	RMSEExt	
1	0.43 ± 0.01	0.46 ± 0.11	0.43 ± 0.03	0.43 ± 0.01	0.46 ± 0.11	0.42 ± 0.03	0.43 ± 0.01	0.46 ± 0.11	0.42 ± 0.03	0.44 ± 0.01	0.46 ± 0.12	0.42 ± 0.03	0.43 ± 0.01	0.44 ± 0.11	0.43 ± 0.03	
2	1.06 ± 0.02	1.18 ± 0.23	1.08 ± 0.14	1.14 ± 0.02	1.26 ± 0.21	1.10 ± 0.12	1.04 ± 0.03	1.26 ± 0.23	1.15 ± 0.12	1.12 ± 0.04	1.18 ± 0.27	1.12 ± 0.12	1.04 ± 0.28	1.22 ± 0.28	1.15 ± 0.14	
3	0.28 ± 0.02	0.52 ± 0.16	0.42 ± 0.06	0.48 ± 0.03	0.79 ± 0.24	0.88 ± 0.15	0.25 ± 0.03	1.17 ± 0.59	0.95 ± 0.26	0.27 ± 0.05	0.60 ± 0.22	0.66 ± 0.13	0.24 ± 0.03	1.61 ± 1.01	1.64 ± 0.86	
4	0.43 ± 0.01	0.46 ± 0.12	0.42 ± 0.03	0.43 ± 0.01	0.46 ± 0.11	0.42 ± 0.03	0.43 ± 0.01	0.46 ± 0.11	0.42 ± 0.03	0.43 ± 0.01	0.46 ± 0.12	0.42 ± 0.03	0.43 ± 0.01	0.48 ± 0.11	0.42 ± 0.03	
5	1.06 ± 0.03	1.25 ± 0.23	1.01 ± 0.14	1.13 ± 0.03	1.26 ± 0.27	1.04 ± 0.13	1.03 ± 0.03	1.28 ± 0.27	1.03 ± 0.13	1.12 ± 0.04	1.24 ± 0.26	1.09 ± 0.11	1.03 ± 0.03	1.24 ± 0.22	0.98 ± 0.13	
6	0.35 ± 0.01	0.48 ± 0.14	0.40 ± 0.07	0.36 ± 0.01	0.46 ± 0.12	0.43 ± 0.06	0.33 ± 0.02	0.47 ± 0.14	0.47 ± 0.08	0.30 ± 0.04	0.48 ± 0.15	0.48 ± 0.10	0.33 ± 0.02	0.51 ± 0.17	0.46 ± 0.09	
7	0.27 ± 0.02	0.50 ± 0.14	0.41 ± 0.08	0.46 ± 0.02	0.73 ± 0.25	0.77 ± 0.18	0.38 ± 0.02	0.83 ± 0.37	0.77 ± 0.16	0.24 ± 0.03	0.59 ± 0.21	0.70 ± 0.10	0.24 ± 0.03	0.93 ± 0.43	1.04 ± 0.39	
8	0.25 ± 0.02	0.48 ± 0.14	0.41 ± 0.06	0.40 ± 0.01	0.55 ± 0.14	0.44 ± 0.08	0.42 ± 0.01	0.65 ± 0.18	0.59 ± 0.10	0.27 ± 0.05	0.52 ± 0.14	0.55 ± 0.08	0.35 ± 0.03	0.45 ± 0.12	0.46 ± 0.06	
9	0.25 ± 0.02	0.45 ± 0.14	0.44 ± 0.08	0.44 ± 0.01	0.70 ± 0.21	0.70 ± 0.11	0.71 ± 0.03	0.66 ± 0.23	0.90 ± 0.13	0.29 ± 0.05	0.54 ± 0.24	0.57 ± 0.11	0.36 ± 0.03	0.46 ± 0.12	0.51 ± 0.07	
10	0.27 ± 0.02	0.46 ± 0.15	0.39 ± 0.06	0.54 ± 0.02	0.81 ± 0.26	0.80 ± 0.12	0.34 ± 0.01	0.74 ± 0.30	0.75 ± 0.11	0.25 ± 0.03	0.58 ± 0.19	0.70 ± 0.18	0.28 ± 0.02	0.57 ± 0.38	0.74 ± 0.31	
11	0.27 ± 0.02	0.50 ± 0.16	0.43 ± 0.06	0.41 ± 0.02	0.69 ± 0.23	0.77 ± 0.14	0.69 ± 0.02	0.79 ± 0.31	0.77 ± 0.17	0.27 ± 0.03	0.72 ± 0.20	0.64 ± 0.11	0.31 ± 0.03	1.00 ± 0.56	0.77 ± 0.38	
12	0.31 ± 0.01	0.46 ± 0.12	0.39 ± 0.06	0.52 ± 0.01	0.67 ± 0.21	0.60 ± 0.09	0.38 ± 0.01	0.62 ± 0.20	0.55 ± 0.08	0.36 ± 0.08	0.71 ± 0.33	0.60 ± 0.26	0.38 ± 0.02	0.45 ± 0.13	0.45 ± 0.07	
13	0.31 ± 0.01	0.48 ± 0.14	0.40 ± 0.05	0.51 ± 0.01	0.64 ± 0.21	0.60 ± 0.09	0.53 ± 0.01	0.59 ± 0.19	0.56 ± 0.07	0.37 ± 0.09	0.76 ± 0.49	0.60 ± 0.19	0.38 ± 0.02	0.47 ± 0.15	0.45 ± 0.07	

From these 11 high-performing models, RF model 3 was selected for further investigation as it affords the understanding of the interaction space by means of the C × P cross-terms. It is also interesting to note that PLS, RR and BLR provided poor performance for model 3 with the exception of SVM, which provided moderately good performance but owing to its black box nature and its lower performance level than RF it was not selected for further study. Another observation worth noting is the unexpectedly high performance afforded by models 1 and 4, which are comprised of the C and C × C descriptor blocks, respectively. This point will be subjected to a further scrutiny in the forthcoming analysis of the prediction scatter plot (Fig. 5).

Figure 4 Plots of the PCA scores (A) and loadings (C) for the 10 aromatase inhibitors and the PCA scores (B) and loadings (D) for the 22 aromatase variants.

Each dot in sub-plots (A and B) represents an aromatase inhibitor and aromatase variant, respectively, while each dot in sub-plots (C and D) represents substructure fingerprint count descriptors and z-scale descriptors, respectively.

Figure 5 Plot of the experimental versus predicted pIC50 values for 13 models built using RF.

Blue circles represent internal sets while the red circles correspond to external tests.

The reliability of the PCM models can be statistically assessed from the differences between the goodness of fit and the predictive ability that is the most reliable models were those for which R2 was not greater by 0.2–0.3 units than Q2. This is because a higher margin of difference between R2 and Q2 is indicative of chance correlation or overfitted models either due to outliers or irrelevant descriptors. Analysis of the R2 − Q2 revealed that RF model 3 was well below the mentioned range indicating its suitability for further analysis.

External validation is an important process for assessing the predictive ability of PCM models. As can be seen in Table 2, external validation of RF model 3 afforded QExt2=0.90±0.03. Thus, it is apparent that external validation for RF yielded a superior performance and were thus subjected to further rigorous validation. Particularly, the PCM models built using RF were further validated using LOCO and LOPO cross-validations to evaluate their ability to extrapolate and predict the inhibitory activities on previously unseen compounds and aromatase variants, respectively. Table 4 summarizes the comparison of the prediction performance of the training set, 10-fold CV and external sets along with the LOPO and LOCO sets. It can be seen that model 3 performed well on both LOCO and LOPO CV sets with QLOPO2=0.89±0.06 and QLOCO2=0.89±0.06, respectively.

Table 4 Summary of the predictive performance of PCM models built using RF.

In addition to the typical internal and external validations, models were also evaluated by leave-one-protein-out (LOPO) and leave-one-compound-out (LOCO) cross-validations.

Model	Training	10-fold CV	External	LOCO-CV	LOPO-CV	
	RTr2	RMSETr	QCV2	RMSECV	QExt2	RMSEExt	QLOCO2	RMSELOCO	QLOPO2	RMSELOPO	
1	0.87 ± 0.00	0.43 ± 0.01	0.86 ± 0.12	0.46 ± 0.11	0.93 ± 0.01	0.43 ± 0.03	0.88 ± 0.06	0.45 ± 0.10	0.89 ± 0.05	0.45 ± 0.09	
2	0.35 ± 0.02	1.06 ± 0.02	0.25 ± 0.22	1.18 ± 0.23	0.25 ± 0.22	1.08 ± 0.14	0.22 ± 0.17	1.15 ± 0.17	0.21 ± 0.16	1.16 ± 0.16	
3	0.95 ± 0.01	0.28 ± 0.02	0.84 ± 0.14	0.52 ± 0.16	0.90 ± 0.03	0.42 ± 0.06	0.89 ± 0.06	0.46 ± 0.08	0.89 ± 0.06	0.46 ± 0.08	
4	0.88 ± 0.01	0.43 ± 0.01	0.85 ± 0.12	0.46 ± 0.12	0.93 ± 0.01	0.42 ± 0.03	0.89 ± 0.06	0.45 ± 0.09	0.88 ± 0.057	0.45 ± 0.09	
5	0.32 ± 0.03	1.06 ± 0.03	0.18 ± 0.19	1.25 ± 0.23	0.33 ± 0.12	1.01 ± 0.14	0.22 ± 0.17	1.17 ± 0.18	0.21 ± 0.17	1.17 ± 0.18	
6	0.93 ± 0.01	0.35 ± 0.01	0.85 ± 0.11	0.48 ± 0.14	0.90 ± 0.04	0.40 ± 0.07	0.88 ± 0.07	0.45 ± 0.10	0.88 ± 0.07	0.44 ± 0.11	
7	0.96 ± 0.01	0.27 ± 0.02	0.83 ± 0.15	0.50 ± 0.14	0.89 ± 0.05	0.41 ± 0.08	0.88 ± 0.07	0.44 ± 0.10	0.89 ± 0.06	0.44 ± 0.10	
8	0.96 ± 0.01	0.25 ± 0.02	0.83 ± 0.15	0.48 ± 0.14	0.89 ± 0.05	0.41 ± 0.06	0.88 ± 0.06	0.45 ± 0.10	0.88 ± 0.06	0.45 ± 0.10	
9	0.96 ± 0.01	0.25 ± 0.02	0.85 ± 0.12	0.45 ± 0.14	0.89 ± 0.04	0.44 ± 0.08	0.89 ± 0.07	0.44 ± 0.12	0.88 ± 0.068	0.44 ± 0.12	
10	0.96 ± 0.01	0.27 ± 0.02	0.84 ± 0.15	0.46 ± 0.15	0.90 ± 0.04	0.39 ± 0.06	0.89 ± 0.06	0.46 ± 0.08	0.89 ± 0.06	0.46 ± 0.08	
11	0.96 ± 0.01	0.27 ± 0.02	0.85 ± 0.12	0.50 ± 0.16	0.88 ± 0.04	0.43 ± 0.06	0.88 ± 0.06	0.46 ± 0.10	0.88 ± 0.06	0.46 ± 0.10	
12	0.94 ± 0.01	0.31 ± 0.01	0.86 ± 0.11	0.46 ± 0.12	0.90 ± 0.04	0.39 ± 0.06	0.89 ± 0.06	0.44 ± 0.10	0.89 ± 0.06	0.44 ± 0.10	
13	0.94 ± 0.01	0.31 ± 0.01	0.86 ± 0.11	0.48 ± 0.14	0.90 ± 0.04	0.40 ± 0.05	0.89 ± 0.06	0.44 ± 0.10	0.88 ± 0.07	0.44 ± 0.11	

A closer inspection of the scatter plot of experimental versus predicted pIC50 values provided in Fig. 5 revealed that models containing the C descriptor block and the C × C cross-terms (models 1 and 4, respectively, as shown in panels 1 and 4 of Fig. 5) had irregular cluttered distribution of the data points. Particularly, data points appear to converge to the same value on the Y-axis planar while being distributed on the X-axis planar. This is because predictions were made against the same compound that has different protein binding partners (i.e., aromatase variants) where each compound-protein pair have its own unique pIC50 values. For example, the same compound tested against different aromatase variants were predicted to have the same pIC50 values. The origin of this problem arose from the fact that only compound descriptors were used for prediction in the absence of protein descriptors. On the other hand, the distribution of the data points in all other models were uniformly and randomly dispersed. In summary, superficially the prediction values seems to be on par with the selected PLS models but a closer look revealed that the underlying data points had predictions converging to the same pIC50 value.

Y-scrambling was performed for 50 times to assess the possibility of chance correlations for the 13 PCM models. Scatter plots of R2 versus Q2 are shown in Fig. 6 for the Y-permutated data set. It can be seen that almost all PCM models (i.e., except for models 2 and 5) had the actual X–Y pair distinctly separated from the scrambled X–Y pairs. Of particular note, RF model 3 afforded distinct separation between the actual and scrambled X–Y pairs. Thus, the model was considered to be robust for further interpretation on the underlying feature importance.

Figure 6 Y-scrambling plots of pIC50 as obtained from PCM models after feature selection.

Mechanistic interpretation of PCM models

Analysis of the feature importance can provide a better understanding on the underlying features that may strongly contribute to the aromatase inhibitory activity. As previously mentioned, RF model 3 was selected as the best model and is thus used for mechanistic interpretation of the feature importance. The efficient and effective built-in feature importance estimators of the RF method was utilized to identify informative features. In general, two measures (i.e., the mean decrease in the Gini index and the mean decrease in prediction accuracy) are used for ranking important features. Because the mean decrease in the Gini index is reported to be robust when compared with the mean decrease in accuracy (Calle & Urrea, 2011), therefore the mean decrease in the Gini index was used to rank features. To avoid possible bias due to random seed of a single data partition, the mean and standard deviation values of the Gini index was calculated from the aforementioned 100 data partitions.

Figure 7 Plot of feature importance for RF model 3.

High Gini index values are indicative of important descriptors.

Figure 7 shows the plot of the Gini index as obtained from RF model 3. A holistic look at the compound component of the top-ten C × P cross-term revealed that important substructures can be classified into two major categories comprising of functional group substructures (e.g., heteroaromatic, alcohol dialkylether moieties and vinylogous carbonyl or carboxyl derivative) and charge-related fingerprints (e.g., “salt”, charge, hetero N basic no H and hetero N non basic). As for the protein component, it was found that the top-ten cross-terms were all described by the z3 descriptor that corresponded to the polarizability and charge properties of amino acids predominantly at positions I133 and I395 while to a lesser extent at the C124 position. The number of occurence(s) of these features in the top-ten descriptors were 3, 6 and 1 for I133, I395 and C124, respectively. This results indicated that polarizability and charge properties of aforementioned isoleucine and cysteine residues may play crucial role in the compound-protein interaction. In the forthcoming paragraphs, a more in-depth look on the interplay between various substructure fingerprints with the two key residues highlighted by the top-ten features from the importance plot will be discussed below.

Three cross-term descriptors pertaining to the interaction of compound substructures to I133 have been identified from the feature importance plot. It should be noted that the description of the characteristics of the substructure fingerprints were obtained by using the SMARTS pattern of the substructure to query the SMARTSviewer (Schomburg et al., 2010) web server.

Firstly, the most important feature was SubFPC184_p133z3 (15.36 ± 4.45) which represents the interaction of the heteroaromatic substructure with the polarizability and charge property at position 133. It can be observed that the heteroaromatic substructure is both electron-rich as well as containing a hydrophobic ring and may thus accomodate hydrophobic interaction at the active site. This is supported by the work of Bansal et al. (2012) where several steroidal AIs based on the 3-keto-4-ene steroidal analogs were synthesized and their findings indicated that the heteroaromatic pyridine ring were the most potent one. Similarly, Khodarahmi et al. (2015) utilized quantum mechanical/molecular mechanical (QM/MM)-based docking to investigate the how compounds interact with aromatase and stressed that the necessary hydrophobic interactions between aromatase and its inhibitors are facilitated via heteroaromatic rings. This feature reflects the binding mechanism by which ligands with the heterocyclic aromatic ring coordinates the azole moiety to the heme iron at the active site while also forming π–π interactions with F221, W224 and I133 and hydrophobic interactions with W224, V369 and T310. The third most important feature was SubFPC16_p133z3 (10.59 ± 2.01), which corresponds to the interaction of the dialkylether substructure with the polarizability and charge property at position 133. Dialkylether refers to a functional moiety having two R groups linked by an oxygen atom. Compounds having this linker group have been reported to afford aromatase inhibitory activities with IC50 values in the range of 5–20 µM (Cantón et al. 2008). The seventh most important feature is SubFPC299_p133z3 (7.65 ± 1.24), which corresponds to the interaction of the salt substructure with the polarizability and charge property at position 133. The substructure descriptor refers to both anions (i.e., atoms with a charge in the range of −1 and −7) and cations (i.e., atoms with a charge in the range of +1 and +7) and thus generally imply that compounds with charged structures are important for interaction.

The second important feature is SubFPC12_p395z3 (10.69 ± 1.76), which corresponds to the interaction of the alcohol substructure with the polarizability and charge property at position 395. Interestingly, compounds having alcohol as a functional group were used in combating endocrine therapy resistance. Cadoo, Gucalp & Traina (2014) claimed that cell cycle regulatory processes play an important role in the development of breast cancer resistance and showed that compounds having alcohol as functional groups namely palbociclib are promising candidates for targeting endocrine therapy resistance. The fourth and fifth most important features are SubFPC299_p395z3 (9.03 ± 1.69) and SubFPC296_p395z3 (8.13 ± 1.50), which corresponds to the interaction of the salt substructure with the polarizability and charge property at position 395 for the former and interaction of the charge substructure with polarizability and charge property at position 395 for the latter. Both substructure descriptors appear to share similar properties where the former defines the salt descriptor as a functional moiety pertaining to anions (i.e., atoms with a charge in the range of −1 and −7) and cations (i.e., atoms with a charge in the range of +1 and +7) while the latter is defined as atoms having either negative or positive charge. Generally, these descriptors indicated that charged structures are important for interaction. The sixth most important feature is SubFPC180_p395z3 (7.87 ± 1.29), which corresponds to the interaction of hetero N basic no H with polarizability and charge property at position 395. Hetero N basic no H can be defined as an aromatic nitrogen atom with neutral charge that is bonded to 3 further total connection and has no bound hydrogen atom. Albrecht et al. (2011) stressed that compounds containing conjugated systems (i.e., N-fused heteroaromatic compounds) are considered to be important privileged structures in drug discovery with notable examples such as zolpidem (i.e., hypnotic properties) and alpidem (i.e., anxiolytic properties). This may therefore indicate that the nitrogen-containing rings are indeed a privileged substructure important for the inhibitory property against aromatase. The eighth most important feature is SubFPC135_p395z3, which corresponds to the interaction between the carboxyl derivative substructure with the polarizability and charge property at position 395. Carboxyl derivatives represents a group of functional moieties comprising of carboxylic acids, esters, anhydride, acyl chloride, amide and nitrile. Antoon et al. (2011) selected sphingosine kinase-2 (Sphk2) from the MAPK pathway as a therapeutic target for treatment of endocrine therapy–resistant breast cancer and suggested ABC294640 (3-(4-chlorophenyl)-adamantane-1-carboxylic acid) as a novel and selective Sphk2 inhibitor. The tenth most important feature is SubFPC181_p395z3, which corresponds to the interaction of hetero N non-basic substructure with the polarizability and charge property at position 395. The hetero N non-basic refers to aromatic nitrogen atom with 2 further total connections or aromatic N with a charge of +1, with 3 further total connections. Nitrogen-containing ring structures are found in FDA-approved drugs targeting aromatase comprising of anastrozole, letrozole and vorozole, which represents the current standard-of-care compounds for preventing the relapse of breast tumor (Ma et al., 2015).

Conclusions

Computational approaches for predicting the activities of AIs can facilitate drug discovery efforts by saving cost and time. The continual increase in breast cancer prevalence has led to the necessity for discovery of novel compounds with strong inhibitory properties towards aromatase. To consider possible effects of aromatase on different AIs, we present a PCM study on aromatase inhibitory activity for discerning the key interaction of AIs with key residues inside binding sites. By utilizing the feature importance estimator of RF, we found that substructures pertaining to charged structures, heteroaromatic rings and alcohol moieties were important features for predicting governing aromatase inhibitory activity. These findings may aid in the design of novel compounds for inhibiting aromatase.

Supplemental Information

Data S1 Supplementary information and supplementary tables describing the data set used in this study

Click here for additional data file.

The authors would like to thank the peer reviewers and Academic Editor for providing comments that helped to improve this manuscript.

Additional Information and Declarations

Competing Interests

Author Contributions

Data Availability

The authors declare there are no competing interests.

Saw Simeon performed the experiments, analyzed the data, wrote the paper, prepared figures and/or tables, reviewed drafts of the paper.

Ola Spjuth, Maris Lapins and Sunanta Nabu analyzed the data, wrote the paper, reviewed drafts of the paper.

Nuttapat Anuwongcharoen analyzed the data, wrote the paper, prepared figures and/or tables, reviewed drafts of the paper.

Virapong Prachayasittikul wrote the paper, reviewed drafts of the paper.

Jarl E.S. Wikberg contributed reagents/materials/analysis tools, wrote the paper, reviewed drafts of the paper.

Chanin Nantasenamat conceived and designed the experiments, analyzed the data, contributed reagents/materials/analysis tools, wrote the paper, prepared figures and/or tables, reviewed drafts of the paper.

The following information was supplied regarding data availability:

The data set used in this study is provided in Data S1.

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
