# Peer review of "Origin of aromatase inhibitory activity via proteochemometric modeling"

_PeerJ, doi:10.7717/peerj.1979_

## Round 0.1 · original submission · Major Revisions

Please heed the reviewer comments provided and find an annotated copy (by reviewer 1) of your manuscript attached to their review. Reviewer 2 has some specific comments about the experimental design that require addressing. Also, an attempt should be made to reduce the statistical descriptors, explain some of the statistics-specific jargon and overall reduce the amount of text dedicated to statistical methodology and analyses. However, I would like to add that the statistical descriptions should remain of sufficient detail for a fellow expert and also, maintain the more 'laymen ' explanations of the goal/purpose of each of the analyses (in methods particularly). This so non-stats experts at least know why each of the analyses is done, even if they do not understand exactly how the analyses are done.

Throughout manuscript:

Verify that genes and gene expression product mRNA are italicized (e.g. CYP19), and that gene expression protein product (e.g. aromatase enzyme) is not. Check document for grammar: for example, the authors often use plural when it pertains to only one subject or object. Specific example: ''The first generation of non–steroid inhibitors was aminoglutethimide,...'' line 402-403.
Present and past tense are used intermittently and inappropriately.

Minor Specifics:

The authors describe only two of the three steps of aromatase activity on lines 31-34.
Line 404: specificity - should be - selectivity.
Trademark names Arimidex and Femara are defined twice in the text - delete one of the mentions.

·

Basic reporting

The paper describes an in depth statistical analysis of the interaction of various AI compounds with variants of the aromatase protein. This is a statistically dense paper and this somewhat detracts from the biological significance of the work.

The work describes the construction of statistical models involving the interaction of substrates with key AA residues and as such is novel and innovative.

Experimental design

The work is clearly described and the statistical approach is described in detail

Validity of the findings

no comments

Additional comments

The paper is very statistically dense - not surprisingly given the nature of the work. However all the statistics tends to distract from the biochemistry of the system. The MS should be reviewed and edited with the aim of making the topic more 'approachable' by thinning out the statistical discussion.

I have made additional comments and a few editorial suggestions in the attached PDF.

Reviewer 2 ·

Basic reporting

The paper titled “Origin of aromatase inhibitory activity via Proteochemometric modeling” focused on aromatase inhibitory activity. In order to model the interaction space of protein and compound interaction, PCM modeling was adopted in this work. This work might help in the design of novel compounds that can inhibiting aromatase and address the issue of aromatase inhibitor resistance. However, several existed issues make this a preliminary work which need revision.

Several typing and grammar error need to be corrected.

Experimental design

1.The standard PCM modeling requires compound descriptors, protein descriptors and interaction cross-terms. Author choose the widely used z-scale protein descriptors and PaDEL compound descriptors, also, cross-terms were designed as multiplication of compound and protein descriptors. However, the results in both table 1,2 and 3 showed that only use compound descriptors can achieves a prefect performance(Also can be found that any model contains compound information can achieves a good results, but P and P*P cannot). This means, only use compound information without any protein information, this model can still be used to predict bioactivity values? I really doubt that. The author needs to explain this carefully.

2.Above issues may also be addressed by the fact that bias of the partitioning of training and validation dataset. Why split the training and testing data as 80%/20% by K-means? The author should clearly state the reasons.

3.After feature selection, it can be found that author give up PLS and adopt ridge regression and random forest. Why the authors choose those two algorithms other than PLS? Are multiplication-based cross-terms suitable in those two algorithms? The authors need to discuss about this.

Validity of the findings

No Comments

Reviewer 3 ·

Basic reporting

1. Reorganize the numbers in the authors and affiliations list in order to have consecutive numbers (line 3).

2. Replace the subtitle "Materials and Method" for "Materials & Methods" (line 59).

3. The acknowledgements section should not be used to acknowledge funders – that information will appear in a separate Funding Statement on the published paper.

4. Use the sentence case format for the subtitles e.g. "Data set" instead of "Data Set" (line 60). "Feature selection" instead of "Feature Selection" (line 109).

5. The residue I474 should be showed in Figure 3 (line 70).

6. "Of the 503 amino acids, 13 amino acid positions were found to be mutated in the investigated variants, as shown in Figure 3" (line 90), there are only 12 highlighted residues in Figure 3.

7. In the legend of Figure 6, "Y" should be in normal style instead of bold.

.

Experimental design

No Comments.

Validity of the findings

No Comments.

Additional comments

This manuscript examines the use of a proteochemometric modeling approach to study the inhibitory mechanisms of AIs towards aromatase. They predicted useful AI substructures that can be useful for designing new AIs with pharmaceutical purposes. In addition, an interesting description of key features in the aromatase binding site for protein-ligand interaction is presented. In general, the experimental design is appropriate and the work is well conducted. However, minor corrections are needed.

---

## Round 0.2 · accepted · Accept

We thank you for an excellent revision on a complex topic that will benefit several cross-science disciplines.

Thomas

Reviewer 2 ·

Basic reporting

No more comments.

Experimental design

No more comments.

Validity of the findings

No more comments.

Additional comments

No more comments.